

# Physical activity level and physical fitness in subjects with chronic musculoskeletal pain: a cross-sectional study

Gabriel Rojas[1,2] and Ignacio Orozco-Chavez[2]

[1] Master in Clinical Exercise Physiology, School of Kinesiology, Universidad Mayor, Santiago, Metropolitana, Chile
[2] Department of Human Movement Sciences, Faculty of Health Sciences, Universidad de Talca, Talca, Maule, Chile

## ABSTRACT

**Background:** Low physical activity (PA) levels and low physical fitness (PF) have been reported in subjects with temporality-based chronic pain; however, it is unknown whether there are differences in subjects with nociplastic pain (NP) compared with subjects with non-nociplastic pain (NNP).
**Objective:** The aim was to compare the levels of PA and PF in patients with chronic, nociplastic, and non-nociplastic musculoskeletal pain.
**Methods:** This is an analytical, cross-sectional study. The sample comprised 30 patients receiving ambulatory physiotherapy treatment. Pain was classified as NP or NNP according to the International Association for the Study of Pain categorization system. The PA level was measured with the International Physical Activity Questionnaire–Short Form, and the PF level was measured with the hand grip strength test (HGS) to assess upper limb strength, the five Repetition Sit-to-Stand Test (5R-SRTS) to assess lower limb strength and power, and the YMCA 3 Min Step Test (YMCA-3MST) to estimate peak $VO_2$. The results were compared with independent samples t-tests (with $p < 0.05$ considered significant). Cohen's d was calculated to determine the effect size.
**Results:** The NP group reported a significantly lower PA level than the NNP group, specifically the vigorous PA ($p = 0.0009$), moderate PA ($p = 0.0002$), and total PA ($p = 0.005$) dimensions. The NP group also showed significantly lower 5R-STS ($p = 0.000$) and HGS ($p = 0.002$) results compared with the NNP group. There were no significant differences in the YMCA-3MST between the NP and NNP groups ($p = 0.635$).
**Conclusion:** It is possible that the neurophysiological and neuromuscular changes related to NP are associated with a reduced ability to perform vigorous PA. Clinicians should identify the presence of NP comorbidities in conjunction with the diagnosis when establishing the therapeutic goals.

Corresponding author
Ignacio Orozco-Chavez,
iorozco@utalca.cl

## INTRODUCTION

Chronic pain is one of the most common conditions for which adults seek medical attention (*Schappert & Burt, 2006*) and is considered a disease by the International Classification of Diseases. Indeed, it is a priority health problem with a prevalence of close to 20.4% (*Dahlhamer et al., 2018*; *Bilbeny, 2019*) and an incidence of close to 8% per year (*Mills, Nicolson & Smith, 2019*). It has been associated with physical problems such as severe restrictions of mobility and activities of daily life, and has been associated to psychological factors as depression, anxiety and catastrophisation (*Gureje et al., 1998*; *Smith et al., 2001*; *Cohen, Vase & Hooten, 2021*).

Pain seems to affect the levels of physical activity (PA)—any bodily movement produced by skeletal muscles that requires energy expenditure—and physical fitness (PF)—the ability to carry out daily tasks with vigor and alertness, without undue fatigue and with ample energy to enjoy leisure-time pursuits and to meet unforeseen circumstances (*Caspersen, Powell & Christenson, 1985*). This becomes extremely important, since PA has numerous benefits in preventing chronic health conditions such as diabetes, hypertension, or cancer (*Warburton, Nicol & Bredin, 2006*), and according to World Health Organization is one of the five leading global risks for mortality. Additionally, PA moderate both the physical and psychological dimensions of pain, and has been recognized an essential non-pharmacological strategy in the management of chronic pain (*Alzahrani et al., 2019*).

Previous studies have found lower levels of PA in subjects with chronic pain (>3 months) compared with asymptomatic subjects (*Griffin, Harmon & Kennedy, 2012*; *Parker et al., 2017*; *Alzahrani et al., 2019*; *Ezeukwu et al., 2019*). Moreover, lower levels of PA are recognized as one of the most important risk factors in subjects with this condition (*Rezaei et al., 2021*). Similarly, subjects with degenerative pathologies in the lumbar region have lower PF, characterized by worse performance in lower limb strength and power (*Staartjes & Schröder, 2018*) and cardiorespiratory resistance functional tests (*Nielens & Plaghki, 2001*) compared with controls without pain. Recently, *Moseng et al. (2014)* compared the PA level by using the International Physical Activity Questionnaire–Short Form (IPAQ-SF) and the PF level by using the 6-Min Walk Test and the 30-s Sit-to-Stand Test between subjects with musculoskeletal conditions and healthy controls. They observed a lower level of vigorous PA and worse performance in clinical tests in subjects with musculoskeletal pathology. However, these studies have categorized pain based on symptom's temporal characteristics, assuming that chronic pain has a duration greater than 3–6 months (*Russo & Brose, 1998*). Moreover, the available studies did not consider the source of the pain or the characteristics of the subjects.

Recently, a reconceptualization of the types of pain was developed considering neurophysiological and clinical aspects, categorizing pain into nociceptive, neuropathic, and nociplastic, and establishing specific therapeutic recommendations for each one. Nociceptive pain is generated by peripheral activation of pain receptors by physical injury or an acute inflammation process. Neuropathic pain is mainly secondary to lesions of the somatosensory system (*i.e.*, carpal tunnel syndrome). Nociplastic pain (NP), on the other

hand, is not fully explained by tissue damage and implies changes in the central nervous system (central sensitization) that amplify the pain signal (hypersensitivity phenomena) usually accompanied by comorbidities such as sleep disturbances, allodynia, hypersensitivity, general fatigue, fear avoidance beliefs, and even cognitive alterations, all of which lead to functional limitations (*Chimenti, Frey-Law & Sluka, 2018*). Thus, a long-lasting pathology categorized as "chronic," such as non-specific low back pain or epicondylalgia, could present pain with nociceptive or nociplastic characteristics.

Although no studies have measured PA levels in subjects with NP, lower PA levels have been observed in subjects with chronic non-specific pain and have been associated with fear avoidance beliefs in subjects with chronic conditions (>3 months) (*Nelson & Churilla, 2015*). *Larsson et al. (2016)* associated lower PA levels measured with Grimby's Activity Scale with kinesiophobia, but not with pain intensity, in adult subjects with chronic pain. Evidence suggests that fear avoidance beliefs and kinesiophobia promote hypervigilance and lead to increased pain sensation, mainly associated with hyperalgesia, hypersensitivity and allodynia (*Bordeleau et al., 2022*), symptoms that determine the presence NP.

In summary, although the influence of chronic pain on the levels of PA and PF is known, studies have only classified subjects' pain and musculoskeletal condition based on their temporality and the old conceptualization of pain. Recently, the International Association for the Study of Pain (IASP) published a decision-making tree and a classification system to categorize the different types of pain according to their clinical presentation, criteria of temporality, the presence of pain comorbidities, and causes (*Nijs et al., 2021*). Thus, the purpose of this study is to compare the levels of PA and PF between subjects with NP and non-nociplastic pain (NNP), classified with the IASP decision-making tree. The hypothesis is that subjects categorized as having NP show a lower level of PA and poorer performance in tests related to PF compared with subjects with NNP. The findings from this study could be of clinical importance given that clinicians should know and understand the contemporary conceptualization of pain in order to carry out biopsychosocial rehabilitation in patients and to integrate individualized strategies of graduated exposure to PA (*Booth et al., 2017*; *Geneen et al., 2017*), and thus impact the quality of life and survival of patients with chronic musculoskeletal pain.

## MATERIALS AND METHODS

### Design

This study used an analytical, cross-sectional design and a convenience non-probabilistic sample. The study measures were carried out at the University of Talca Physiotherapy Clinic and in the Physiotherapy Care Center of Lircay Clinic, Talca, Chile. This study followed the recommendations for the reporting of cross-sectional studies contained in Strengthening the Reporting of Observational Studies in Epidemiology- STROBE (*von Elm et al., 2007*). STROBE checklist can be reviewed in Supplemental file 1.

### Participants

The sample selection process is illustrated in a flow diagram (Fig. 1). Sixty-two potentially eligible subjects were selected from 184 active patients who attended the outpatient

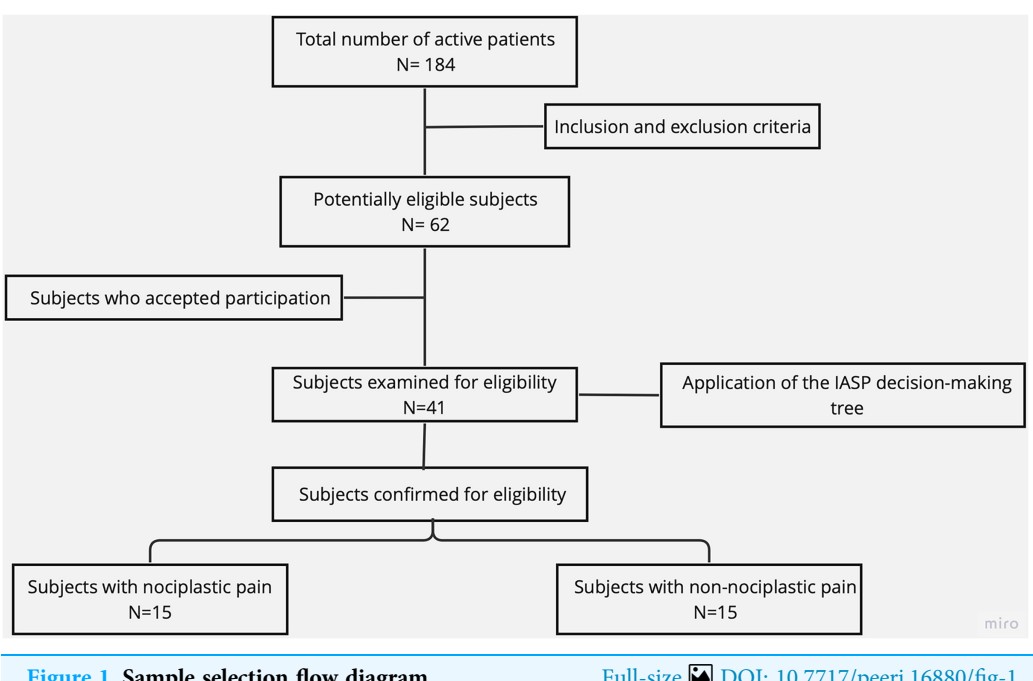

**Figure 1 Sample selection flow diagram.**

rehabilitation centers of the University of Talca and the Lircay Clinic between October and December 2022. Of these, 41 subjects were examined for eligibility based on the IASP clinical decision-making tree (*Nijs et al., 2021*). A sample size calculation for independent groups was performed (power = 0.8, alpha = 0.05) based on the mean and standard deviation of previous studies (*Moseng et al., 2014*), and considering a 50% loss rate. Based on the calculation, 30 participants were randomly recruited for the NP (*n* = 6 men, 9 women) and NNP (*n* = 6 men, 9 women) groups.

The participants included in the study met the following inclusion criteria: (1) age between 30 and 65 years, (2) medical diagnosis of chronic musculoskeletal condition or pathology without surgical intervention, (3) pain lasting ≥3 months, (4) independent walking, and (5) the ability to go up and down stairs independently. The exclusion criteria were: (1) a medical diagnosis of sensory or vestibular alterations, (2) a medical diagnosis of heart disease, and (3) a physiotherapy-patient relationship with the researchers. All the participants read and signed the informed consent form approved by the scientific ethics committee of the University of Talca (approval number 24-2022).

## Measures

### Clinical interview and categorization of nociplastic pain

Subjects examined for eligibility underwent a clinical interview process that included general questions such as age, sex, body mass index (BMI), diagnosis (the compromised anatomical region), and evolution time of the symptoms, and specific questions to detect the type of pain, corresponding to the IASP clinical decision-making tree and classification system (based on; *Nijs et al., 2021*), which was applied by a single blind investigator during the initial evaluation session. This tool contains questions that, according to the answers of

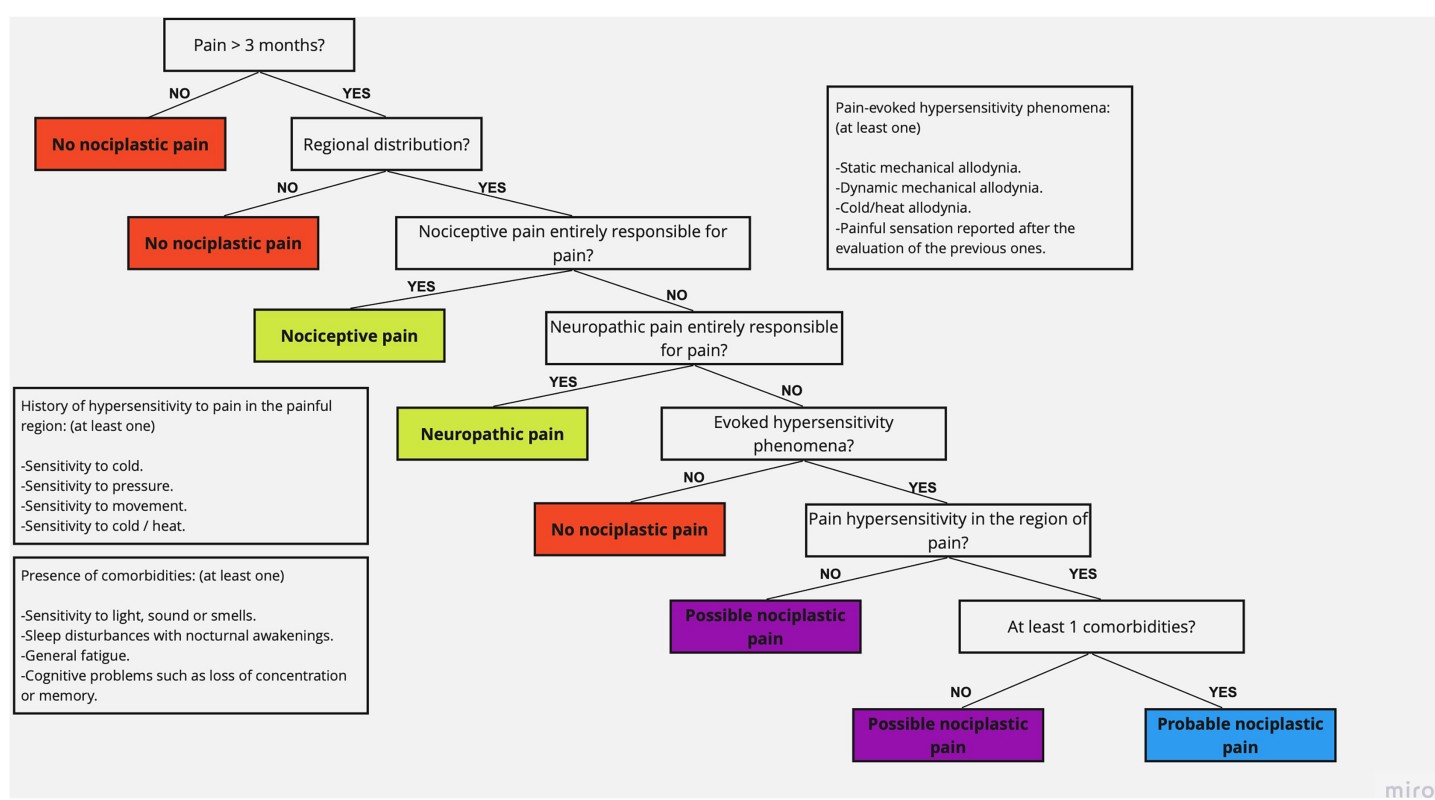

**Figure 2 IASP clinical decision-making tree and classification system (extracted data from *Nijs et al., 2021*).**

the subject, allow the categorization of pain as nociceptive pain, neuropathic pain, or NP (Fig. 2). Moreover, it allows differentiating between *possible* NP (no comorbidities) and *probable* NP (at least one comorbidity) (*Kosek et al., 2021*). The participants were categorized as NNP or NP, regardless of the subcategorization of each one.

### Physical activity level

The validated Spanish versions of the IPAQ-SF (*Roman-Viñas et al., 2010*) was used to measure the frequency and time dedicated to transport and light, moderate, and vigorous PA during the last 7 days. Light activity was considered that which does not imply physical effort or changes in cardiovascular parameters; moderate activity was considered that which produces a moderate increase in ventilation, heart rate, and sweating for at least 10 min (continuous); and vigorous activity was considered that which produces a greater increase in the same variables, for ≥10 min (*Mantilla Toloza & Gómez-Conesa, 2007*). Authorization was requested from the author, who also provided the validated version of the document translated into Spanish. The results are reported as minutes per day for each of the PA dimensions. To categorize the total PA, the self-reported time was converted to the metabolic equivalent (METs): 3.3 for light PA, four for moderate PA, and eight for vigorous PA. For total PA, all the dimensions were added up and the final value is reported as METs/min/week and has been categorized as high, moderate, and low PA levels according to previous studies (Table 1).

**Table 1 Physical activity categorization criteria.**

| Physical activity level | Condition |
| --- | --- |
| Low | Do not perform PA or perform it but it does not reach the moderate and high categories. |
| Moderate | 5 or more days of any combination of light, moderate, or vigorous PA that reaches 600 METs-min/week |
| High | 3 or more days of vigorous physical activity or accumulating 1,500 METs-min-week |
| | 7 or more days of any combination of light, moderate, or vigorous physical activity that reaches 3,000 METs-min/week |

### Physical fitness

PF was characterized by performing clinical tests that measured lower limb strength and power, hand grip strength (HGS), and cardiorespiratory fitness (CRF). All the tests were performed in the same visit, and a rest time of 5 min was considered between tests.

The 5-Repetition Sit-to-Stand test (5R-STS) was performed to measure the strength and power of the lower limbs (*Staartjes & Schröder, 2018*; *Alcazar et al., 2018*, *2020*). It includes the use of a standard chair (48 cm high) leaning against a wall. The participant had to position her-/himself sitting, with his/her hands crossed on his/her chest, and from that position stand up completely and sit down again a total of five times. Time was measured in seconds. If the subject did not complete the test or if it took more than 30 s, then the result was reported as 30 s (*Jones et al., 2013*).

To assess the strength of the upper limbs, the HGS test (*Köklü et al., 2016*) was performed with a CAMRY® brand digital dynamometer. The subject held the dynamometer firmly with the hand to be tested in a bipedal position, in line with the forearm at the thigh level, separated from the body. The subject was asked to squeeze the dynamometer as hard as possible for 3 s. The test was repeated twice in each hand to obtain the highest value, adding both hands to obtain the final HGS value in kilograms (*Pescatello, Riebe & Thompson, 2014*). The dominant hand was measured first, and alternate attempts were made between the dominant and non-dominant hand to allow each upper limb to rest.

CRF was assessed by measuring peak oxygen consumption using the YMCA 3-Min Step Test (YMCA 3MST) (Van *Kieu et al., 2020*). After a 2-min rest period while sitting, the subject was asked to step up and down a 30 cm high step 72 times (steps) for 3 min (the ascent and descent of both feet was considered a step), at a 24 step/min rhythm established by a digital metronome set to 96 beats/min (4 beats = 1 step). The subject had to stop at the end of the time, sit, and remain still in order to measure the heart rate with a digital saturometer 5 s after finishing the test. Peak $VO_2$ was obtained using an equation formulated by the Korean Institute of Sports Sciences and validated by *Chung et al. (2014)*:

**Men: $VO_{2peak} = 70.597 - 0.246 \times (Age) + 0.077 \times (Height) - 0.222 \times (weight) - 0.147 \times (Heart\ Rate)$**

**Women: $VO_{2peak} = 70.597 - 0.185 \times (Age) + 0.097 \times (Height) - 0.246 \times (weight) - 0.122 \times (Heart\ Rate)$**

Each participant performed the tests in the order stated above. There was no warm-up prior to the tests because it is not included in the protocols of the tests used in this study.

## Statistical analysis

The Shapiro-Wilk test was used to determine whether the variables followed a normal distribution. The mean and standard deviation are reported for the total IPAQ-SF survey (METs/min/week), the 5R-STS (s), the HGS test (kg), and the YMCA 3MST (peak $VO_2$ [mL/kg/min]). An independent samples t-test was used to compared between the groups. Finally, Cohen's d (*Cohen, 2013*) was calculated to determine the effect size of the differences. Statistical analyzes were performed using SPSS Version 26.0 software (IBM Corp., Armonk, NY, USA). The statistical significance level was set at $p < 0.05$.

## RESULTS

Table 2 shows the NP and NNP group characteristics. Based on the Shapiro-Wilk test, the data for both groups were normally distributed. Independent samples t-tests showed no significant differences in BMI ($p = 0.438$) and symptom evolution time ($p = 0.052$) between the groups, but there was a significant difference in age ($p = 0.002$). Specifically, the NP group had a mean age of 49.6 years while the NNP group had a mean of 40.1 years. Regarding the diagnosis by anatomical region, 33.3% of the subjects with NP presented upper limb diagnoses, while 26.6% of the subjects with NNP had upper limb diagnoses. Lower limb diagnoses accounted for 26.6% and 53.3% of the subjects with NP and NNP, respectively. Finally, trunk and spinal diagnoses corresponded to 40% and 20% of the subjects with NP and NNP, respectively.

## Physical activity level

The NP group reported a significantly lower level of vigorous PA compared with the NNP group ($p = 0.0009$): 0.0 and 645.87 METs/min/week, respectively. In the moderate PA dimension, the NP group reported 0.0 METs/min/week, significantly lower ($p = 0.0002$) than the NNP group, which reported 348.0 METs/min/week. In the light PA dimension, there was no significant difference between the groups ($p = 0.08$). Total PA was significantly lower for the NP group ($p = 0.005$): 652.86 METs/min/week compared with 1486.66 METs/min/week for the NNP group. Considering the PA level categorization, more subjects with NP (53.3%) were characterized at the low PA level, while the NNP group had the highest proportion of subjects (66.6%) characterized at the moderate level (Table 3).

## Physical fitness

The NP group had a significantly longer execution time in the 5R-STS test ($p = 0.000$), taking 6.94 s longer (65%) than the NNP group; the effect size was large (d = 1.87). In addition, the NP group presented significantly lower HGS ($p = 0.002$), generating 20.43 kg (54.8%) less than the NNP group; the effect size was large (d = 1.28). Finally, the difference in the estimated peak $VO_2$ between the NP and NNP groups was 1.05 mL/kg/min. This difference was not significant ($p = 0.635$), and the effect size was small (d = 0.17) (Table 4).

**Table 2 Sample characteristics.**

| Subjects characteristics | NP group (n = 15) Mean (SD) | NNP Group (n = 15) Mean (SD) | p-value |
|---|---|---|---|
| Age (years) | 49.6 (8.1) | 40.1 (8.2) | 0.002 |
| BMI (kg/cm2) | 28.1 (3.5) | 28.2 (2.4) | 0.438 |
| Symptoms evolution (Months) | 8.40 (4.5) | 5.67 (2.6) | 0.026 |
|  | n (%) | n (%) |  |
| Diagnosis (Anatomical region) Upper Limb Lower Limb Spine | 5 (33.3) 4 (26.6) 6 (40.0) | 4 (26.6) 7 (53.3) 3 (20.0) |  |
| IASP Comorbidities Sleep Deprivation (SDp) General Fatigue (GF) Both D and GF | 5 (33.3) 4 (26.6) 6 (40.0) |  |  |

Note:
NP, Nociplastic pain; NNP, Non-nociplastic pain; SD, Standard deviation; BMI, Body max index; IASP, International Association for the Study of Pain; SDp, Sleep Deprivation; GF, General Fatigue.

**Table 3 Physical activity level categorization.**

| IPAQ-SF Score METs min/week | NP Group (n = 15) Mean (SD) | NNP Group (n = 15) Mean (SD) | p-value |
|---|---|---|---|
| Vigorous activity | 0.00 (0.00) | 645.87 (652.56) | 0.0009 |
| Moderate activity | 0.00 (0.00) | 348.00 (292.94) | 0.0002 |
| Light activity | 653.46 (214.45) | 492.80 (378.15) | 0.08 |
| Total | 652.86 (213.53) | 1,486.66 (986.01) | 0.005 |
| **Physical activity level** | n (%) | n (%) |  |
| Low | 8 (53.3) | 3 (20) |  |
| Moderate | 7 (46.6) | 10 (66.6) |  |
| High | 0 (0) | 2 (13.3) |  |

**Table 4 Physical fitness of the participants during 5R-STS, HGS y YMCA 3MST.**

|  | NP Group (n = 15) Mean (SD) | NNP Group (n = 15) Mean (SD) | Mean difference (IC 95%) | p-value | Cohen'd |
|---|---|---|---|---|---|
| 5R-STS (s) | 17.52 (4.81) | 10.58 (2.06) | 6.94 (4.11 a 9.77) | 0.000 | 1.87 |
| HGS (Kg) | 37.26 (12.95) | 57.69 (18.52) | 20.43 (8.47 a 32.38) | 0.002 | 1.28 |
| YMCA 3MST ($Vo_2$) | 37.70 (6.88) | 38.75 (4.90) | 1.05 (3.42 a 5.51) | 0.635 | 0.17 |

Note:
5R-STS, Five Times Sit-to-stand Test; HGS, Hand Gripp Strength Test; YMCA 3MST, 3 Min Step Test; Vo2, Peak Oxygen Consumption in ml/kg/min.

## Discussion

The present study aimed to compare the levels of PA and PF in subjects with NP and NNP. The main results showed that compared with the NNP group, the NP group reported a significantly lower level of PA—specifically the vigorous PA, moderate PA, and total PA dimensions of the IPAQ-SF—and also presented worse PF, characterized by significantly lower upper limb strength and lower limb strength and power. There was not a significant difference in peak $VO_2$ between the NP and NNP groups.

The PA level was measured using the IPAQ-SF, which estimates this value based on the amount of time each week that an individual dedicates to vigorous, moderate, or low activity. This information was used to the total METS of PA. In this study, the NP group reported a lower level of moderate, vigorous, and total PA compared with the NNP group. This difference may be explained by the characteristics of pain and comorbidities associated with NP symptoms reported by the participants (*Nijs et al., 2021*). The subjected with NP presented pain of non-neuropathic or nociceptive origin as main symptom, with the presence of evoked hypersensitivity phenomena, and reported sleep deprivation ($n = 5$), general fatigue ($n = 4$), or both symptoms ($n = 6$). These comorbidities increase pain sensitivity, which could explain the avoidance of movements that involvepowerful or sustained muscle contractionsand vigorous PA, and lead to a maladaptation of the neuromuscular and cardiovascular systems (*Fitzcharles et al., 2021*).

Based on the literature, subjects with classic chronic pain have much lower PA levels based on the IPAQ-SF compared with healthy controls (*Ezeukwu et al., 2019*). However, no studies have categorized pain as nociplastic or non-nociplastic according to the IASP criteria. Previous studies have related PA to pain comorbidities, avoidance behaviors, and central sensitization symptoms in subjects with chronic low back pain. *Huijnen et al. (2011)* evaluated the PA level by using the Beacke Physical Activity Questionnaire in subjects with chronic low back pain. The authors categorized the participants according to their behaviors, observing that subjects with avoidance behaviors and fear of movement presented lower levels of self-reported PA. More recently, *Zheng et al. (2023)* compared the PA level measured by accelerometry between subjects with chronic low back pain and with and without symptoms of central sensitization; they observed different PA patterns between both groups. Although total PA was not different between the groups, subjects with central sensitization showed more inactive time compared with subjects without sensitization. While avoidance behaviors and central sensitization could represent an indirect relationship to the clinical categorization of NP, more studies that subclassify pain according to the IASP criteria are necessary to corroborate the results of the present study. Longitudinal studies are required to establish a better relationship between the PA level and NP. In addition, studies are needed that assess the PA level with accelerometry as well as self-reported questionnaires.

The NP group presented a lower PF level compared with the NNP group—specifically, lower limb strength and power based on the 5R-STS test and HGS. The HGS test can be a reliable method for assessing total muscle strength, and indirectly the level of muscle sarcopenia (*Gonçalves et al., 2018*). The 5R-STS has been validated as an objective test to

measure functional impairment related to lower limb power and strength in subjects with various chronic pathologies such as non-specific low back pain (*Klukowska et al., 2021*).

Previous studies have reported that subjects with chronic pain perform worse in PF tests such as the HGS test, the 5R-STS, the 6-Min Walk Test, and the Astrand test, among others (*Choi et al., 2021*; *Klukowska et al., 2021*). These results are consistent with the present findings, which could be explained by neural adaptations of processing and modulation of nervous system signals generated by subjects with NP (*Fitzcharles et al., 2021*; *Chimenti, Frey-Law & Sluka, 2018*). NP is related to central sensitization processes, which implies changes in the representation of sensory and motor areas. The poor differentiation of afferent pain pathways causes an inhibition of motor efferent stimuli, which results in lower muscle recruitment, and decreased ability to generate force and muscle power (*Hodges et al., 2013*). This causes catastrophizing and avoidance behaviors that lead to disuse, disability, and altered neuromuscular activity during functional movements (*Lund et al., 1991*; *Vlaeyen et al., 1995*), contributing to worse test performance, muscle weakness, and lower PF. This could also explain the lower levels of vigorous PA (*i.e.*, aerobics, running, and fast bicycling) reported by subjects with NP, because these require both the recruitment of a greater number of motor units and more muscle groups in a shorter amount of time.

Peak $VO_2$ of the YMCA 3MST test was not significantly different between the NP and NNP groups. Previous studies have reported that subjects with chronic pain have worse cardiorespiratory performance in submaximal and maximal tests compared with healthy controls (*Smeets et al., 2006*; *Duque, Parra & Duvallet, 2009*; *Doury-Panchout et al., 2012*). However, no studies have analyzed differences between different pain subgroups. The absence of difference between the groups in the YMCA 3MST test could be explained by the submaximal characteristics of this test; hence, it may not be the best tool to estimate peak $VO_2$ in subjects with pain. In the present study, the NP and NNP groups did not show differences in activities with a low metabolic cost (light PA) categorized with the IPAQ-SF, which includes tasks such as walking or up and down stairs. If the level of adaptation of subjects with NP and NNP is similar for the task performed, it may not be demanding enough to show differences in the estimated peak $VO_2$. Although direct measurement of $VO_2$ could be an alternative to represent PF, it would not be recommended given the limitation of subjects with NP to perform vigorous PA, as observed with the PA measurements. Due to the characteristics of NP, it would be difficult to determine whether a difference in $VO_2$max is due to their cardiorespiratory performance or to exercise-induced pain. Other representative variables of PF should be considered in future studies.

This study has some limitations that need to be mentioned. The sample size is small relative to the high prevalence of chronic pain. Moreover, this study used a convenience sample of people attending health centers. It is important to increase the sample size and to recruit a random sample. Regarding the data analysis, there was a significant 9.5-year age difference between the NP and NNP groups, which could have influenced the differences found in upper limb strength and lower limb strength and power. However, the literature shows that the decrease in strength in both men and women occurs after the age of 60 years

(*Mancilla, Ramos & Morales, 2016*). Given that the mean age of the entire sample was 44.85 years, age can be ruled out as a confounding variable. Regarding the measurements, using the IPAQ-SF to determine the PA level implies a subject bias as it is self-reported and depends exclusively on the subject's understanding of the tool (*Hagstromer et al., 2010*). Nevertheless, it is widely used to assess PA and has been validated for the Chilean population (*Palma-Leal et al., 2022*). In addition, an indirect peak $VO_2$ test was used in the present study. Although sufficient methodological safeguards were taken to validate the procedure, it is advisable to carry out laboratory tests that allow measuring $VO_2$max objectively, which provides a better characterization of PF. Finally, the present study measured the differences in the levels of PA and PF in subjects with NP and NNP; however, some characteristics of the population must be taken into consideration. First, unlike other studies, a broad spectrum of diagnoses rather than a single pathology (*i.e.* low back pain) was considered. Although this approach does not influence the IASP categorization of NP, future studies should consider a larger sample to allow differentiation between the diagnosis, subtypes of pain, and comorbidities. On the other hand, other psychosocial factors, such as depression (*Codella & Chirico, 2023*) could affect the level of PA The IASP decision-making tree considers specific comorbidities, such as sleep deprivation and general fatigue, ignoring other conditions that may affect the level of PA and PF. It would be recommended that future studies conduct a comprehensive clinical interview and integrate other evaluations associated with pain experiences (*i.e.*, Tampa Kinesiophobia Scale) in order to identify other factors associated with nociceptive pain and central sensitization processes.

## CONCLUSIONS

Subjects with NP reported a lower level of total PA and also had significantly lower upper limb strength and power and lower limb strength than subjects with NNP. These differences could be explained by the presence of hypersensitivity and comorbidities, which have been associated with central sensitization processes and avoidance behaviors in subjects with NP. These findings are relevant for professionals regarding the evaluation and categorization approach in patients with chronic pain. They could consider these findings to provide more individualized interventions when incorporating physical exercise and PA programs with the aim of improving PF in patients, and thus impact chronic diseases and survival of chronic patients. Future research is required to support these results and to establish a clinical difference between subjects with different subtypes of NP.

## ACKNOWLEDGEMENTS

The authors would like to thank the directors of the health centers of the University of Talca and Clínica Lircay for their willingness to facilitate the development of this study. In addition, we thank each of the subjects who voluntarily decided to participate in our evaluations.

### Funding
The authors received no funding for this work.

### Competing Interests
The authors declare that they have no competing interests.

### Author Contributions
- Gabriel Rojas conceived and designed the experiments, performed the experiments, prepared figures and/or tables, authored or reviewed drafts of the article, and approved the final draft.
- Ignacio Orozco-Chavez analyzed the data, prepared figures and/or tables, authored or reviewed drafts of the article, and approved the final draft.

### Human Ethics
The following information was supplied relating to ethical approvals (*i.e.*, approving body and any reference numbers):

The University of Talca Scientific Ethical Committee.

### Data Availability
The raw measurements are available in the Supplemental File.

### Supplemental Information
Supplemental information for this article can be found online at http://dx.doi.org/10.7717/peerj.16880#supplemental-information.

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
