# Peer review of "Physical activity level and physical fitness in subjects with chronic musculoskeletal pain: a cross-sectional study"

_PeerJ, doi:10.7717/peerj.16880_

## Round 0.1 · original submission · Major Revisions

The rationale for the study needs to be better supported in the introduction. The literature used to support their work does not always appear to be pertinent to the primary aim of the study. Both reviewers find merit in your work provided their are substantial revisions to address their concerns.

Reviewer 1 ·

Basic reporting

Abstract:
1. Line 28: Repeating abbreviations (NP and NNP are already addressed in the background sections of the abstract).
2. Line 25: Please use the abbreviations (PA & PC) or delete the abbreviations.
3. Line 39-40: Conclusion of the abstract seems repeating the results again. Please provide the conclusion of the study based on the observed results.
Introduction:
4. Line 45: please revise the following: “it’s considered…”.
5. Line 46: please revise the following: “near to 20,4%”
6. Line 54-60: In order to improve readability, please break down into separate sentences (i.e., address nociceptive, neuropathic, and nociplastic pain separately).
7. Line 54-60: It was mentioned that based on the new approach (i.e., mechanism-based approach), acute pain is recognized as nociceptive pain while chronic pain is recognized as nociplastic pain. Based on the cited paper (Chimenti et al. 2018), nociceptive pain is primarily due to nociceptor activation typically resulting in acute localized pain, whereas nociplastic pain are due to alterations of nociceptive processing. It was suggested that nociplastic pain is typically chronic pain (e.g., fibromyalgia), however, it was also suggested that some conditions can involve both nociceptive and nociplastic pain such as low back pain or knee osteoarthritis (which is known as chronic pain). In other words, some chronic pain conditions can be classified to nociceptive pain.
8. Line 73-74: “physical fitness” is the term that has been widely used to describe “the ability to carry out daily tasks with vigor and alertness….”. Is there any specific reason for using the term “physical condition” instead (might be useful to cite paper if this term has been used”)?
9. Line 81: Please revise (Nielens, 2021) to (Nielens, 2001).
10. Line 82: “International Physical Activity Questionnaire short-form (IPAQ-sf)”
11. Line 72-86: The main purpose of the study is to compare between individuals with nociplastic pain and non-nociplastic pain (both group with pain conditions). However, all cited studies are addressing the difference between chronic pain individuals and healthy controls. It might be useful to address some previous studies that compared PA and/or PC between different types of pain conditions.
Results:
12. Line 187-195: Is it possible to provide what were the actual pain conditions of participants (e.g., fibromyalgia, low back pain, etc.).
13. Table 1: Please revise the symptoms evolution (months) for NP group “8,40”.
14. Line 204-207: Please provide the reference for the PA level categorization.

Experimental design

Methods:
15. Line 107: Please address (in method or in result) how many females and males were assigned into NP and NNP group.
16. Line 138-145: Please explain why self-reported PA was converted to the METs, instead of reporting minute/day for light, moderate, vigorous.
17. Line 137-142: Please provide more detail on each PA domain (e.g., what was the criteria for light, moderate, vigorous PA?).
18. Line 138-145: Which criteria was used to categorize as high, moderate, and low PA levels (i.e., what was the actual values for the categorization)? Please provide reference.
19. Line 148-173: Please provide more detail on physical fitness tests. Were all tests completed in one single visit? What was the order of the tests (e.g., 5R-STS followed by HGS and YMCA 3MST)? Did the participants perform any types of warm-up protocol before the tests?
20. Line 156-161: For HGS test, how long was the contractions? How long was the rest period between attempts? Which arm was tested first (e.g., dominant arm or right arm, etc.)?
21. Line 169-173: Please provide the reference for the equation.

Validity of the findings

Discussion & Conclusion:
22. Line 232: Please clarify what PN symptoms stand for.
23. Line 247-248: It was mentioned that the subgroup of subjects with the lowest PA could correspond to NP and the subgroup with the highest PA could correspond to NNP. Based on the cited paper (Parker et al. 2017), it was mentioned that 2 out of 12 individuals showed higher number of steps than many of the matched controls. This seems like a very weak evidence to suggest that these 2 individuals are with NNP and other 10 individuals are with NP.
24. Line 277: Please clarify what do you mean by “which are related to strength and power tasks…”.
25. Line 273-276: Please double-check and revise the sentence.
26. Line 286: It was suggested that submaximal test (YMCA 3MST) may not be the best tool to obtain peak VO2 in subject with pain. Further, it was mentioned that the direct measurement of VO2 or activities of a moderate/high level of physical activity is suggested. If the reason why NP group had no vigorous/moderate activity was observed in the present study was due to those factors you addressed (e.g., increased pain, central fatigue, etc.), using submaximal test might be the only option to determine their VO2. In other words, if higher intensity exercise was used to determine Vo2 between NP and NNP, it is difficult to determine if the difference in Vo2 was due to their cardiorespiratory performance or due to exercise-induced pain.
27. Line 320: Please provide explanation of CF.
28. The main purpose was to compare between NP and NNP individuals and the authors observed differences in some variables between two groups. However, throughout the discussion, there is a lack of explanations of these findings. Discussing the potential reasons to explain the differences between NP and NNP is recommended.

Additional comments

The present study aims to compare the physical activity level and physical fitness between individuals with nociplastic pain and non-nociplastic pain. The topic by itself is interesting, however, the rationale behind (e.g., logic and fluency) is not strongly supported in the introduction. Furthermore, it is recommended to further provide/discuss the explanations of the findings of the study in the discussion section.

Reviewer 2 ·

Basic reporting

In general, the introduction to the study shows the context with relevant and well-referenced literature. The general structure of the text follows the journal's standards; it has a coherent context and justification. It explains the problem's magnitude, the topic's relevance, and the study's need. However, it has some limitations, which must be addressed for the manuscript to be considered for publication.

1. The English writing should be proofread entirely
2. Some references should be updated, especially regarding demographic or epidemiological aspects.
3. Some introduction arguments should be revised, rewritten, and supplemented to improve the readers' understanding and strengthen the study's implications.
4. According to STROBE Statement, authors should check the items that should be included in the report cross-sectional studies.

Experimental design

The research is original and within the scope of the journal. The research question, inferred from the objective and the hypotheses, is well-defined. It has clinical relevance and responds to an identified knowledge gap. However, the description of the methods must be improved so that the research is replicable and with a high technical standard.

Validity of the findings

Although the study is novel because there are no previous data specifically related to this type of pain (nociplastic), there are doubts regarding the validity of the findings, mainly because there is a lack of clarity in the description of the recruitment process, selection of the participants and assignment to study groups. The data provided is statistically sound but needs to be completed regarding identifying potential biases that were not identified, measured, and reported. The conclusions are linked to the research question and are limited to supporting results.

Annotated reviews are not available for download in order to protect the identity of reviewers who chose to remain anonymous.

---

## Round 0.2 · Major Revisions

The reviewers found that the manuscript was substantially improved. However, Reviewer 2 suggested additional discussion was necessary to better characterize some study limitations.

Reviewer 1 ·

Basic reporting

no comment

Experimental design

no comment

Validity of the findings

no comment

Additional comments

In the revised manuscript, the authors satisfactorily addressed all my concerns/comments, and improved the quality of the manuscript.

Reviewer 2 ·

Basic reporting

In general, the writing was substantially improved in each of its items. The English wording was revised. The general context was improved, with updated bibliographic references sufficient to support the arguments. The tables and figures have been prepared appropriately, and the raw data have been shared.

Abstract: Abbreviations and concepts were improved.

Line 22: "This analytical, cross-sectional study was performed according to the STROBE checklist" should be replaced with "This is an analytical, cross-sectional study."
If the authors followed the recommendations for reporting cross-sectional STROBE studies, the methodology could describe this.


Introduction (Line 101-104): The introduction was substantially improved, the references were updated, and the authors improved the justification. However, you can still strengthen the main outcomes' implications for the subjects. This study aims to compare the physical activity levels and physical performance in subjects with chronic pain, nociplastic and non-nociplastic types. It is an interesting and relevant topic, considering that chronic pain is a highly prevalent disease that limits mobility and, therefore, can impact the level of activity and physical condition of the subjects in any of its categories. A low level of physical activity is related to lower physical performance. It is considered an important health indicator linked even to the risk of mortality (according to the WHO, people with an insufficient level of physical activity have a risk of death between 20% and 30% higher compared to people who achieve a sufficient level of physical activity). Hence, health conditions that favor a low level of PA require special attention. In this sense, the need to strengthen the study's implications was explained in a previous comment. Through the written introduction, discussion, and conclusion, the authors mainly reinforce the need to categorize the type of pain in the subjects for a differentiated intervention. However, the implications and impact of low physical activity and physical performance on this subject and the need for intervention are not addressed. It is suggested to reinforce these implications and the future perspectives based on the differences found.

Experimental design

The study adequately meets the technical and ethical level. The research methods were corrected and described in greater detail. However, it is suggested that the details of the comorbidity associated with the group with nociplastic pain be incorporated into the sample characterization table (since these data are only available in the raw data). They are elements that the authors consider in their discussion and conclusion to explain the findings found and would allow better visualization for the reader.

Line 110-111: "This study was performed according to the STROBE checklist (Supplementary file 1) for observational studies" should be replaced by: The study followed the recommendations for the reporting of cross-sectional studies contained in Strengthening the Reporting of Observational Studies in Epidemiology- STROBE.

Validity of the findings

The study is replicable. All underlying data that have been provided are robust. The authors improved the description of recruitment, selection of participants, and assignment to study groups. However, the levels of physical activity and physical fitness can be modulated multi-dimensionally. For example, pain can manifest with physical and psychological symptoms, affecting Physical activity levels and performance differently. In this sense, although the pain characteristics and associated comorbidities were considered in the nociplastic pain group (sleep deprivation, general fatigue), there are other psychosocial factors that, together with pain, could affect physical activity levels, for example, depression. This study did not include a characterization of psychosocial factors or other comorbidities, which could be a study limitation and should be discussed.

---

## Round 0.3 · accepted · Accept

The authors addressed the reviewers' comments.

Reviewer 2 ·

Basic reporting

No comment

Experimental design

No comment

Validity of the findings

No comment

Additional comments

No comment